# Effectiveness of Lifestyle Health Promotion Interventions for Nurses: A Systematic Review

**DOI:** 10.3390/ijerph17010017

**Published:** 2019-12-18

**Authors:** Natalia Stanulewicz, Emily Knox, Melanie Narayanasamy, Noureen Shivji, Kamlesh Khunti, Holly Blake

**Affiliations:** 1School of Applied Social Sciences, De Montfort University, Leicester LE1 9BH, UK; 2Infant Nutrition and Metabolism, University of Granada, 52005 Granada, Spain; Emily_knox2@hotmail.co.uk; 3School of Health Sciences, University of Nottingham, Nottingham NG7 2RD, UK; melanie.narayanasamy@nottingham.ac.uk (M.N.); holly.blake@nottingham.ac.uk (H.B.); 4School of Primary, Community and Social Care, Keele University, Staffordshire ST5 5BG, UK; n.shivji@keele.ac.uk; 5Diabetes Research Centre, University of Leicester, Leicester LE1 7RH, UK; kk22@le.ac.uk; 6NIHR Nottingham Biomedical Research Centre, Nottingham NG7 2UH, UK

**Keywords:** lifestyle intervention, systematic review, nurses, employee wellbeing, workplace intervention

## Abstract

Background: Prior research has investigated various strategies to improve health, wellbeing and the job-related outcomes of nurses. However, the scope of this evidence is not clear and the types of intervention most likely to have positive outcomes are unknown. Objective: To provide an overview and synthesis of the effectiveness of interventions conducted with the goal of improving health, wellbeing and the job-related outcomes of nurses. Methods: A systematic database search was conducted from January 2000 to December 2018, with pre-defined criteria (Cochrane Central Register of Controlled Trials; MEDLINE and PubMed; EMBASE; CINAHL; PsycINFO; and BioMed Central). In total, 136 intervention studies with a total sample of 16,129 participants (range 9–3381) were included and evaluated. Data extraction, quality assessment and risk of bias analyses were performed. Results: Studies included randomised controlled trials (RCTs; n = 52, 38%), randomised crossover design studies (n = 2, 1.5%) and non-randomised pre-post studies with a control group (n = 31, 23%) and without a control group (n = 51, 37.5%). The majority of interventions focused on education, physical activity, mindfulness, or relaxation. Thirty-seven (27%) studies had a multimodal intervention approach. On average, studies had relatively small samples (median = 61; mode = 30) and were conducted predominantly in North America (USA/Canada, n = 53). The findings were mixed overall, with some studies reporting benefits and others finding no effects. Dietary habits was the most successfully improved outcome (8/9), followed by indices of body composition (20/24), physical activity (PA) (11/14), and stress (49/66), with >70% of relevant studies in each of these categories reporting improvements. The lowest success rate was for work-related outcomes (16/32). Separate analysis of RCTs indicated that interventions that focus solely on education might be less likely to result in positive outcomes than interventions targeting behavioural change. Conclusions: Interventions targeting diet, body composition, PA, or stress are most likely to have positive outcomes for nurses’ health and/or wellbeing. The methodologically strongest evidence (RCTs) is available for body composition and stress. Interventions relying solely on educational approaches are least likely to be effective. Organisational outcomes appear to be more challenging to change with lifestyle intervention, likely requiring more complex solutions including changes to the work environment. There is a need for more high-quality evidence since many studies had moderate or high risk of bias and low reporting quality.

## 1. Introduction

Nurses are at the frontline of public health and spend considerable time promoting healthy lifestyle behaviours to patients and their families. However, studies of lifestyle behaviours in nurses have typically shown a pattern of non-adherence to public health guidelines around physical activity (PA), sedentary behaviour (SB), diet, smoking and alcohol consumption [1,2,3,4,5,6,7,8]. Overweight and obesity have been found to be significantly higher amongst nurses than other healthcare professionals and those working in non-health-related occupations [9]. A national survey showed that 25% of English nurses are obese (BMI: body mass index ≥30.0), with obesity rates higher than those for other healthcare professionals [9]. Obesity increases the risk of diseases including diabetes, heart disease, osteoarthritis and cancer [10], and increases the risk of musculoskeletal (MSK) problems. MSK is a leading cause of sickness absence [9,11] and is prevalent in nurses [12,13], but could be improved with lifestyle changes such as exercise [14]. A recent secondary analysis of nationally representative cross-sectional data from the United Kingdom (UK) has suggested an upward trend in the health-related behaviours of nurses relative to the general working population. This showed improved habits relating to smoking, fruit/vegetable intake, and physical activity (PA), but not for alcohol consumption; although overall adherence to public health guidelines remains inadequate [15].

Nurses’ knowledge about healthy lifestyle behaviours does not necessarily result in healthier lifestyle behaviours [7], and lifestyle choices outside of the workplace (e.g., low levels of leisure-time PA) are not necessarily compensated for by the nature of the job role (e.g., nursing work is predominantly comprised of light-intensity PA) [16].

There can be many barriers to engagement in healthy lifestyle choices within the nursing working environments. These include lack of access to exercise facilities [17], barriers to healthy eating practices due to adverse work schedules, individual barriers, and aspects of the physical workplace environment and social eating practices [18].

Mental ill-health is a serious concern in the nursing profession [19,20,21,22]; it is one of the leading causes of sickness absence in the UK National Health Service (NHS), incurring a significant financial burden to healthcare services [11]. Rates of work-related stress, emotional exhaustion and burnout are high [23,24], and the prevalence of depression may be higher in nurses than in the general population [25]. There are many complex organisational issues that may impact on nurses’ mental wellbeing and care quality (e.g., staffing shortages and workload, turnover, failure to retain staff, and shift patterns). However, there is potential for healthier working environments offering lifestyle interventions to improve factors such as stress levels, job satisfaction, and retention of the nursing workforce, which ultimately may improve the quality of care being provided.

Both, the health and wellbeing of nurses impact significantly on healthcare organisations. In addition to the physical and mental health of workers, there is a great financial burden on healthcare organisations in the form of sickness absenteeism [26], and presenteeism (working while sick) [27,28]. Nurses are four times more likely to exhibit presenteeism compared to other occupations [29], with presenteeism costing twice as much as sickness absenteeism [30]. Nurses’ ill-health impacts on productivity [31], care quality [31,32], absenteeism and turnover [33], continuity of care [34] and patient safety (e.g., through increased patient falls, medication errors and staff-to-patient disease transmission) [29,35,36]. Links have been made between unhealthy lifestyle behaviours, stress, work engagement and job satisfaction [37,38,39,40]. Nurses perceive that being overweight reduces their work performance [41] and influences their willingness to promote health to others [42,43,44]. Moreover, obese nurses have suggested that the public may be less likely to trust their health promotion messages [43]. This shows that nurses’ own lifestyle and health behaviour choices may impact on care quality, and ultimately, patient clinical outcomes.

Systematic reviews on health promotion interventions addressing both individual (including physical and psychological health outcomes) and organisational outcomes of working-age nurses have not yet been conducted. Chan and Perry [45] published a similar review including intervention studies published up to 2011, but focused only on individual health outcomes. Other published reviews have focused on one specific outcome and/or have restricted the focus to one specific type of nursing job role [46,47,48]. The number of included articles in some existing reviews is low, the findings are mixed, the quality of research in the reviews is low to moderate and the full range of lifestyle interventions for nurses is not well described. Furthermore, the effectiveness of interventions for registered nurses is not always clear since some studies combine healthcare professionals with student samples (e.g., [49]), despite differences between these demographics that may influence their health, lifestyle choices, attitudes and behaviours.

There is a strong rationale, therefore, for the provision of services and facilities within healthcare organisations to support nurses’ health and wellbeing. However, there is a need to determine *which* interventions are likely to be most effective in improving individual outcomes (i.e., lifestyle behaviour; physical and psychological health) and organisational outcomes (i.e., employee engagement, job satisfaction, performance, productivity, sickness absence, patient safety and care) in order to inform decisions about the provisions to be offered through health and wellbeing programmes.

### Aims

The primary aim of this systematic review was to provide a synthesis of the literature on lifestyle interventions for nurses, and to establish the efficacy of interventions intended to improve behavioural health risk factors and/or behavioural or clinical outcomes of working-age nurses. The secondary aim was to identify the efficacy of these interventions in improving work-related outcomes.

## 2. Materials and Methods

This systematic review was undertaken in line with the guidance for reviews in health care [41], and was registered in PROSPERO on 29th May 2018 (CRD42018098642). The protocol of this review was published on 25th May 2019 in the Journal of Nursing and Practice [50]. Reporting was guided by the PRISMA checklist [51].

### 2.1. Eligibility Criteria

#### 2.1.1. Types of Studies

This review included only original studies, consisting of randomised and non-randomised controlled trials (RCTs, non-RCTs) and non-controlled intervention studies. Non-controlled intervention studies could include before-and-after cohort studies, or interrupted time series studies. Studies with no intervention, or studies with an intervention but reporting no data, were excluded. 

#### 2.1.2. Types of Participants

The studies included working-age nurses. In publications with mixed participant groups, we included studies where nurses constituted at least 50% of the target population. Studies primarily targeting student nurses, retired nurses, healthcare assistants (HCA), other unqualified nursing assistants, or other occupational groups were excluded.

#### 2.1.3. Context/Setting

Any workplace setting in which nurses are accessed. Studies from any country were included.

#### 2.1.4. Types of Interventions

Behavioural and/or educational lifestyle interventions, either alone or in combination, were included, which were aimed at improving any of (but not limited to) individual health risk factors, clinical health and psychological health (as specified in Section 2.1.5 below). Interventions therefore primarily targeted nurses’:Health risk factors: overweight or obesity, diet, PA, smoking habits, problem drinking.Clinical health: type 2 diabetes, stroke, chronic heart disease, cancers, hypertension.Psychological health: work-related stress, mood, self-efficacy.

We excluded interventions that focused solely on health and safety initiatives, or the improvement of clinical skills (e.g., hand washing or infection control methods, patient moving and handling techniques and nurse-patient communication skills). Interventions were excluded that focused primarily on the treatment of psychological disorders (such as post-traumatic stress), although wellbeing interventions focused on stress-management and the prevention of psychological disorders or compassion fatigue were included (e.g., studies focused on nurses’ stress or personal wellbeing in which factors such as compassion fatigue, communication skills or psychological disorder are measured as outcomes or are targeted as part of a multicomponent lifestyle intervention). Interventions that focused solely on organisational changes without a focus on individual health and wellbeing were excluded. Lifestyle interventions primarily targeting nurses’ physical, clinical and/or psychological health, but measuring work-related outcomes, were included.

#### 2.1.5. Comparator(s)/Control

Since the purpose of the review was to identify lifestyle health promotion interventions, we did not define comparator(s)/control in advance. For the controlled trials identified, we described the interventions the control group received. In studies for which no alternative interventions were used for the control group, this was stated. For any non-controlled studies identified, comparator(s)/control was not applicable.

#### 2.1.6. Types of Outcome Measures

The studies assessed outcomes either as changes in health knowledge, health behaviours, disease risk factor indices, related mortality and morbidity or changes in organisational outcomes (including job-related factors, patient safety and care). Changes to relevant health risk factors, clinical health outcomes, psychological health outcomes and organisational outcomes were specified. This includes scores from baseline to last available follow-up. Outcome measures include:

##### Health Risk Factors:

Changes to weight, BMI, waist or other anthropometric indices, changes to diet (e.g., intake of fruit and vegetables, lipid and cholesterol levels), changes in levels of PA (e.g., frequency, duration, intensity), changes to smoking habits (e.g., number of cigarettes smoked per day, cessation attempts), and changes in alcohol consumption.

##### Clinical Health Outcomes:

Related morbidity, hypertension with changes in systolic and/or diastolic values, type 2 diabetes with changes in incidence prevalence or indices of glycaemic control such as HbA1c. Longer-term related morbidity or mortality including incidence of acute coronary syndrome, renal or liver failure, peripheral vascular disease, cerebrovascular disease, incidence of neurovascular complications from type 2 diabetes, and cancers.

##### Psychological Health Outcomes: 

Measures of stress, anxiety, depression, burnout, compassion fatigue (CF), self-efficacy.

##### Work-Related or Organisational Outcomes: 

Measures of job satisfaction, organisational commitment, employee engagement, sickness absenteeism, early retirement or intentions, performance, productivity, staff retention rates, staff turnover rates, patient safety and care.

### 2.2. Search Strategy

Seven electronic databases were searched (using MeSH and free text search terms) for eligible studies including the Cochrane Central Register of Controlled Trials, MEDLINE and PubMed, EMBASE, CINAHL, PsycINFO, and BioMed Central from January 2000 to December 2018 (MN, NSh). The reference lists of the identified records and relevant reviews were checked (HB, EK). Only studies published in English were included. The search strategy is included in Table 1.

### 2.3. Selection Processes

Three reviewers (MN, NSh, EK) independently performed a study selection process and any duplicated records were removed. The titles and abstracts of the remaining records were screened, and full texts were sought for records which clearly referred to behavioural and/or educational lifestyle interventions for working age nurses. The full texts were then assessed for eligibility, taking into account intervention type, study population, outcomes reported, and language. The agreement on inclusion and exclusion was reached through discussion between the reviewers (MN, NSh, EK), with any disagreements resolved by a fourth reviewer (HB).

### 2.4. Data Extraction

The data extraction was performed independently by two reviewers (NSt, HB) and agreement was reached through discussion (i.e., both reviewers checked the data extraction table, and discussed any inconsistencies to reach a consensus; this was needed if the data were not particularly clear). The details on participants, setting, intervention, and outcome measures were extracted from each study. The methodological features of all studies were assessed using the CONSORT checklist [52,53].

### 2.5. Risk of Bias

Three reviewers independently reviewed and critiqued the retrieved papers (NSt, HB, EK) and any disagreements were discussed. The risk of bias was assessed using the Cochrane Handbook classification [54] for all included papers. The risk of selection, performance, detection, attrition and reporting bias were assessed. The risk of bias for non-RCT studies was assessed as high for the categories that could not be satisfied in such designs.

### 2.6. Method of Synthesis

The studies were summarised narratively, which is acknowledged as an appropriate approach to take when assessing data from heterogenous study designs [55]. We report narrative synthesis of the findings from the included studies, structured around the type of intervention, target population characteristics, type of outcome and intervention content. We also provide summaries of the intervention effects for each study (see Appendix A).

## 3. Results

### 3.1. Included Studies

The initial search resulted in 17,126 potential articles. A review of the titles and abstracts resulted in a sample of 567 being selected for further review. The abstracts and full texts of these papers were compared against the inclusion and exclusion criteria, which resulted in 435 articles being excluded. The remaining 132 papers were hand searched. The hand search resulted in four additional papers being included. Subsequently, a total of 136 papers were included in this review. Figure 1 demonstrates the flow of the study selection process.

#### Designs of the Included Studies

The studies in this review included randomised controlled trials (RCTs; n = 52), randomised crossover design studies (n = 2), non-randomised pre-post studies with a control group (n = 31) and without a control group (n = 51). They all examined or compared interventions aimed at improving physical or mental health and wellbeing and/or work-related outcomes in nurses. Comparison groups included a wait-list control (n = 12), an active control (n = 27), no intervention (n = 41), or care as usual (n = 5). Fifty-one studies had no control/comparison group. Appendix A describes the characteristics of the included studies.

### 3.2. Characteristics of the Samples

The sample sizes in the included studies ranged from nine to 3381 participants (total participants = 16,129; median 61; mode 30). The mean age of participants in the studies reporting this value was 39.48 years (SD = 7.18); 37 studies did not provide such data. In the majority of studies in which gender was reported (n = 101), females, on average, accounted for 91% of participants, with 23 studies reporting all female nurses. In terms of geographical location, 58 studies were conducted in North or South America (including 53 from USA/Canada), 39 in Asia, 30 in Europe, eight in Australia, and one study used a cross-cultural sample from both America and Asia. Only two studies were conducted in the UK.

### 3.3. Characteristics of the Interventions

The included studies were grouped according to intervention type. Many of the intervention studies included more than one intervention type (n = 37 studies).

#### 3.3.1. Educational Interventions

Educational interventions were the most common. A total of 58 studies [55,56,57,58,59,60,61,62,63,64,65,66,67,68,69,70,71,72,73,74,75,76,77,78,79,80,81,82,83,84,85,86,87,88,89,90,91,92,93,94,95,96,97,98,99,100,101,102,103,104,105,106,107,108,109,110,111,112] focused on education (excl. smoking education; see below for smoking cessation interventions) or had education as an element of a more complex or combined intervention. Education provision typically related to coping with stress (n = 19) [55,59,60,65,69,72,75,76,84,85,86,91,92,94,95,96,102,106,110], emotion regulation (n = 8) [56,73,74,90,99,100,105,109], communication skills (n = 2) [56,89], positive thinking/positive intervention (n = 4) [79,101,103,107], searching for meaning (n = 1) [108], compassion fatigue/burnout (n = 8) [66,70,71,80,81,82,87,93], the selection optimisation compensation (SOC) model (n = 2) [57,62], self-care (n = 5) [61,63,77,82,97], healthy lifestyles (n = 7) [67,75,83,88,104,111,112] and prevention of back pain/body posture (n = 5) [58,64,68,78,98]. The education was delivered using digital platforms (e.g., websites (e.g., [55,67,104]), apps (e.g., [103,113]), email [92]), but also via one-to-one sessions (e.g., [73,112]), and group educational sessions or workshops (e.g., [56,57,58,59,60,61,62,63,64,65,66,68,69,70,71,72,73,74,75,76,77,80,81,82,83,84,85,86,87,88,89,90,91,93,94,95,96,97,98,99,100,101,102,105,106,107,108,109,110]).

#### 3.3.2. Physical Activity and Dietary Interventions

Thirty-five studies included some form of PA [60,69,75,76,78,83,88,96,98,113,114,115,116,117,118,119,120,121,122,123,124,125,126,127,128,129,130,131,132,133,134,135,136,137]. The PA interventions included walking [83,88,96,113,115,119,123,127,131,133,136], standing [119], aerobics [60,75], aerobics and resistance exercise [118], yoga [116,122,125,126,132,135], Tai-Chi [130], endurance training [117], muscle strength promotion [117,128], stretching [119,134,137], daily exercise [78,121], physiotherapy exercise [76], exercises with equipment (stair-stepper: [114], treadmill, Wii^TM^: [96,115], elastic bands and kettlebells [120,137,138], back muscle exercises [98,124,129], or unspecified [69]). 

Ten studies investigated the effects of interventions based on diet and/or water consumption [75,83,88,96,113,116,119,127,139,140]. The dietary interventions included dietary or healthy eating education [75,83,88,113,116,140], use of diet diaries [127], diet supplement (Omega-acid pills, [139]), cooking sessions [88], healthy snacks [96], or hydration intervention [96,119,127].

#### 3.3.3. Smoking Behaviour

The interventions in three studies were aimed at smoking cessation [141,142,143]. Smoking cessation interventions were delivered using group-based education [142,143], self-directed education [142], or nicotine patches [141].

#### 3.3.4. Mindfulness and Relaxation

Thirty-one studies included a mindfulness [62,66,69,93,126,133,135,144,145,146,147,148,149,150,151,152,153,154,155,156,157,158,159] or meditation [87,160,161,162,163,164,165,166] intervention (most often with the use of mindfulness-based stress reduction (MBSR) as a mindfulness program). Twenty-four studies [63,70,71,72,86,96,105,151,160,162,167,168,169,170,171,172,173,174,175,176,177,178,179,180] included, solely or in combination, some form of relaxation, such as massage [169,170,171,173,175]—with or without aromatherapy, aromatherapy bath for feet [172], guided imagery [70,71,72,151], breathing exercises [63,162], muscle relaxation [86,151,178], Benson’s relaxation technique [174], listening to music and resting [167], playing music [176,179], listening to relaxing texts on CD [96,177], engaging in forms of art [63,105,160,168] such as general art [160], reading poems [63], silk painting [168], dance and mandala painting [105], using a relaxation ball [96], or knitting [180]. 

#### 3.3.5. Other Intervention Types

There were 12 studies that featured other non-medical intervention types. These included complementary and alternative therapies (CAT), such as Reiki [181], touch therapy [182], auriculotherapy [183,184,185], light therapy [186], mantram repetition [187], and neurolinguistic programming (NLP) [188]. Other non-medical intervention types included telephone support groups [189], sleep interventions [177], and occupational health screening and/or consultation [111,190]. 

#### 3.3.6. Intervention Duration and Follow-up

Overall, the intervention length ranged from 10 min (e.g., one short massage session) to 2 years (mean 2.16 months; SD = 2.6; mode 2 months), although six studies did not provide sufficient details on intervention length. The majority of outcomes were assessed immediately after the end of the intervention, with only a few studies assessing medium or longer-term intervention effects.

#### 3.3.7. Intervention Settings

The interventions were predominantly delivered in hospital wards/medical centres or ambulatory clinics (n = 123), with less common settings being a hospice (n = 2) [87,179], and residential or care homes for older people (n = 4) [74,75,98,109], as well as private home care settings [64] (See Appendix A for more details). One study had a sample which included hospital nurses as well as nurses who were municipal employees [69], another study included nurses from various settings (both community and institutional) [189], whereas another four did not specify where the nurses were employed [108,140,142,174].

### 3.4. Measures Used

The outcome variables were assessed by a multitude of measures, and the vast majority of measures were self-report questionnaires. The key questionnaires used are presented below.

#### 3.4.1. Health Risk Factors

[i] *Clinical Health Outcomes*: Self-report measures of general health were more often used and included the Short-Form Survey (SF) [67,130,133,147,168,183,186], the Symptom Checklist (SCL-90) [147,150,157], the General Health Questionnaire (GHQ) [72,100,126,146,160], the Pennebaker Inventory of Limbic Languidness [176], and the Standard Shift-work Index [177].

[ii] *Body Composition:* The most often included measures were BMI [67,83,97,112,113,114,115,116,117,123,128,131], fat mass [97,115,131], and waist circumference [113,116,123,128,131]. Only one study included the measure of dual-energy X-ray absorptiometry [96]. Muscle/joint flexibility/durability was assessed using objective measures (e.g., Sit and Reach Test and others) [114,118,128,130,134,137,138].

[iii] *Diet and Nutrition:* Mostly measured using self-report measures such as the Health-Promoting Lifestyle Profile [HPLP; 61,67], the New South Wales (NSW) Health Survey [127], snack intake (self-report [140]), the Food Frequency Questionnaire [113] and the Rapid Block Food Screener [112]. Two studies measured cholesterol level as an outcome [83,118].

[iv] *Physical Activity and Sedentary Behaviour:* Most often measured by self-report questionnaires including the Health-Promoting Lifestyle Profile [HPLP; 61,67], the Yale Physical Activity Survey [112], the International Physical Activity Questionnaire (IPAQ) [83], and the Active Australia Questionnaire [127]. A small number of studies used objective measures of PA (see below). The objective measures included activity trackers (pedometers [96,112,115,123,131], or accelerometers [113]); the UKK walking test [75], or aerobic capacity using the VO_2max_ test [117,118].

[v] *Smoking Behaviour:* The measures included abstinence [141], number of cigarettes smoked, number of nurses smoking, nicotine dependence, confidence to resist smoking [142], carbon monoxide (CO) level, and smoking cessation status [143].

#### 3.4.2. Psychological Health Outcomes

[i] *Stress and Coping:* Many self-report questionnaires were used, although the most frequently applied was the Perceived Stress Scale (PSS) [59,83,91,106,110,112,130,133,147,154,155,156,160,163,165,170,181,187]. Other measures included the Nursing Stress Scale (NSS) [55,85,86,107,130,148,157], the Coping with Stress Questionnaire [109], the Visual Analogue Scale (VAS) [160,164], the Job-Related Tension Index [160], the Vasconcelos Stress Symptoms List (VSSL) [56,184,185], the Ways of Coping Questionnaire [184,189], the Job Stress Scale [121], the Secondary Stress Symptoms (from ProQoL [71,80,81,135,153,180]), the Questionnaire on Medical Worker’s Stress [122], the Perceived Occupational Stress Scale (POSS [171]), the Stressor Scale for Paediatric Oncology Nurses (SSPON [90]), the Four Dimensional Symptoms Questionnaire (4DSQ [104,190]), the Occupational Stress Inventory [175], the Perceived Stress Questionnaire [92], the Personal and Organisational Quality Assessment [94], the Stress Coping Scale [143], the Expanded Nursing Stress Scale [99,188], the Coping Stress-Revise [132], the Brief Coping Orientation to Problems Experienced Scale [178], the Profile of Mood States [106], and the AIDS Impact Scale [189]. Blood pressure and/or cortisol were used as proxy indicators of stress in just five studies [131,135,167,172,182].

[ii] *Depression and Anxiety*: These outcomes were assessed with the Depression, Anxiety and Stress Scales (DASS) [66,93,146,149,156], the Hospital Anxiety and Depression Scale (HADS) [60,121,139,176], the State-Trait Anxiety Inventory (STAI) [66,164,169,174,178,187], the VAS [164,182], the Generalized Anxiety Disorder Scale (GAD-7) [62,91,97], the Faces Anxiety Scale [171], the Profile of Mood States [172], the Brief Symptom Inventory [143,190], the Patient Health Questionnaire-9 [97], the Centre for Epidemiologic Studies Depression Scale [177], and the Beck Depression Inventory [155].

[iii] *Burnout:* Only four scales were used in the included studies to measure burnout. Specifically, the Maslach Burnout Inventory (MBI) [56,60,63,65,69,72,73,76,82,84,89,94,121,125,135,144,148,155,160,161,189], the Compassion Fatigue Self-Test [158], the ProQoL [70,71,80,81,93,105,145,149,152,153,162,180], and the Copenhagen Burnout Inventory [133,152].

[iv] *Mindfulness:* This outcome was measured with the Mindful Attention Awareness Scale (MAAS) [91,145,148,156,163], the Five Facet Mindfulness Questionnaire [149,158], and the Freiburg Mindfulness Inventory (FMI) [125,153].

[v] *General Wellbeing and Satisfaction:* The following were used: the Psychological Wellbeing Scale [102,161], the Satisfaction With Life Scale [144,149,151,161], the World Health Organisation Questionnaire (WHO-5) [57,62,74,93], the World Health Organization Quality of Life-BREF (WHOQOL-BREF) [62], the Warwick–Edinburgh Mental Wellbeing Scale (WEMWBS) [168], the Cooperation-World Organization of Colleges Academics (COOP/WONCA) [75], the Endicott’s Quality of Life Enjoyment and Satisfaction Short Form [187], the Functional Assessment of Chronic Illness Therapy (FACIT) [87,108,126], the EuroQol [129], the Perceived Wellness Scale [152], and the Subjective Happiness Scale [156].

[vi] *Self-Efficacy*: Self-efficacy was measured using three self-reported measures; the Self-Efficacy Scale [58], the Exercise Self-Efficacy Scale [124], and the Caring Efficacy Scale [150]. 

#### 3.4.3. Work-Related or Organisational Outcomes

These outcomes were assessed using the following measures: the Work Ability Index [57,62,76], the Work Limitation Questionnaire [55,115,130], the Productivity Scale [160], the Nurses Work Functioning Questionnaire [111], the Job Enjoyment Scale [112], the Work Analysis Instrument for Hospitals [57,62], the Job Satisfaction Scale [144], the Scale for Shift-work Complaints [186], the Nurse Satisfaction Scale [55], number of sick days [75,130,135,171], the Caring Efficacy Scale [150], the Nursing Job Satisfaction Scale [82,189], the Job Control and Job Demands Scales [84], presenteeism (Health and Work Performance Questionnaire [139]), the Nurses Work Functioning Questionnaire (NWFQ, [104,190]), Need for Recovery after Work (the Experience and Evaluation of Work Questionnaire, [104,190]), the McCloskey/Mueller Satisfaction Scale [165], the Utrecht Work Engagement Scale (UWES-9, [92,135]), work performance (the Personal and Organisational Quality Assessment [94]), the Quality of Work Life (Brooks and Anderson’s scale [103]), the Team Building Questionnaire [179], the Job Diagnostic Survey [108], and the Benefits of Working (Benefits Finding Scale [108]).

### 3.5. Overall Effect of the Interventions

The majority of interventions in the included studies resulted in significant improvements in at least one measured outcome, although some of the outcomes were not improved following intervention exposure. Health behaviours (including PA, diet, smoking, alcohol consumption), clinical or health outcomes, and work-related outcomes were less often measured than indices of psychological wellbeing. Overall, the strongest evidence was for (i.e., improvements reported in a high number of studies) improvements in stress, anxiety, and burnout (mostly emotional exhaustion (EE) and depersonalisation (DP)). There was some evidence for (i.e., improvements reported in a lower number of studies) personal achievement (PAch), wellbeing, compassion (satisfaction and fatigue), work functioning, PA and indices of body composition (BMI, weight). The outcomes that were less likely to change following intervention were depressive symptoms, personal accomplishment (burnout subdomain), life and job satisfaction, and job control. Based on the outcomes measured in included studies and this overall trend, it appears that lifestyle interventions were more likely to positively influence emotional-based outcomes (heavily relying on mood state, emotional valence), and less likely to positively impact cognitive-focused outcomes (such as quality of life or job-related perceptions, which are assessed more cognitively than emotionally).

### 3.6. Specific Effects of the Interventions

#### 3.6.1. Health Risk Factors

##### Clinical Health Outcomes 

Physical Symptoms and General Health:

Of the included studies, 17 included a measure of general health or physical symptoms.

Of these, 11 [67,72,100,133,136,146,157,160,176,177,183] demonstrated improvements in health following intervention, including physical symptoms ([157] as measured by the Symptom Checklist-90; SCL-90), and physical health ([67,183] as measured by Short Form-36; [72,100,146,160] as measured by the General Health Questionnaire; [133] as measured by Short Form-12; [177] as measured by the Standard Shiftwork Index), psychosomatic symptoms ([176] as measured by the Pennebaker Inventory of Limbic Languidness), sickness and doctor’s visits [160], and perception of one’s health ([136], measured with a single item). Of the 11 studies reporting improvements in a measure of health or physical symptoms, four were RCTs [100,157,176,183], three were non-randomised controlled studies [67,160,177] and four were uncontrolled studies [72,133,136,146]. Those studies demonstrating health improvements reported interventions based on mindfulness ([146,157]—plus yoga), health [67] and coping [72] education, emotional intelligence (EI) education [100], relaxation ([160,176] with meditation), auriculotherapy [183], sleep and relaxation [177] and PA [136]. 

A further six studies reported no significant change in health or physical symptoms [75,126,130,131,147,168], as measured by Short Form-36 [130,147,168], SCL-90 [147], health complaints [75], GHQ [126], or cardiovascular health (i.e., resting blood pressure, [131]). Of the six studies reporting no changes in measures of health or physical symptoms, three were RCTs [75,130,131], and three were non-randomised controlled studies [126,147,168]. The interventions failing to demonstrate positive outcomes used PA [130,131], PA with mindfulness [126], PA with stress education [75], mindfulness [147] and art-based relaxation [168]. None of the studies reported other clinical outcomes. 

##### Body Composition and Functioning

Twenty-four studies included a measure of body composition or body functioning. Outcomes related to at least one measure of body composition (e.g., weight, BMI, waist or other anthropometric indices) improved following intervention in 20 studies [58,76,78,96,97,98,112,114,115,116,117,118,124,128,129,130,131,134,137,138]. These studies mostly assessed changes in weight/BMI/fat mass [96,97,112,114,115,116,131], or body functioning (e.g., flexibility, muscle strength, aerobic capacity, correct body posture, pain reduction [58,76,78,98,114,117,118,124,128,129,130,131,134,137,138]), with some assessing more than one outcome. Of the 20 studies reporting improvements in body composition, 10 were RCTs [58,76,115,118,124,128,129,130,131,138], six were non-randomised controlled studies [96,114,116,117,134,137] and four were uncontrolled studies [78,97,98,112]. Successful interventions predominantly relied on PA (n = 11, [114,115,117,118,124,128,129,130,131,134,137], or PA with education [76,96,98,116,138]; only four used solely education as a mode of intervention [58,78,97,112].

Not all of the studies showed improvements in at least one aspect of body composition; four studies [64,83,113,123] reported no significant improvements in low back pain [64], BMI or waist size [83,113,123]. Of the four studies reporting no improvements in body composition, one was a non-randomised controlled study [64] and three were uncontrolled studies [83,113,123]. Unsuccessful interventions used PA (walking—[123]), education on body mechanics [64], or pedometer challenge with healthy lifestyle education [83,113].

##### Diet and Nutrition

Nine studies reported nutrition (healthy eating) as an outcome, with all but one study [83] reporting positive outcomes for diet or nutrition following intervention [61,67,88,112,113,118,127,140]. Of the eight studies reporting improvements in diet or nutrition, two were RCTs [118,140], two were non-randomised controlled studies [61,67] and four were uncontrolled studies [88,112,113,127]. These interventions were predominantly based on education, including an e-health website [67,112], face-to-face education sessions [88,112], creating self-care plans [61], keeping track of one’s steps and diet [127], providing physical resources (water bottle, sandwich box, healthy cookbook [88,127]), providing cooking classes [88], setting action plans for lower snack intake [140], goal setting for changes in diet [113], with five studies also incorporating a PA element (Wii exercises [112], aerobics [118], or walking [88,113,127]). All the outcomes measured in these seven studies were based on self-report methods, and included reports of fruit and vegetable intake [112,113,127], cholesterol [118], snack intake [140], breakfast consumption [127], Health-Promoting Lifestyle Profile (HPLP) score [61,67] or self-devised [88] questionnaires. The single study that showed no positive outcome was an uncontrolled study, which used blood tests to determine cholesterol level. This unsuccessful intervention combined nutritional intervention with PA. 

##### Physical Activity and Sedentary Behaviour 

Outcomes related to level of PA improved following intervention in 11 studies [67,75,83,88,112,115,118,119,123,127,136]. In these studies, lifestyle interventions increased the frequency of PA (i.e., days walking per week) [88], duration of PA (e.g., steps walked, number of sessions, or minutes/hours per day/session) [88,112,115,123,127], intensity of PA (e.g., light, moderate, vigorous) [127], kilocalories (Kcal) burnt per week [112], and awareness of one’s activity (i.e., stretching, walking, standing [119]). Most of the studies used self-report questionnaires (e.g., HPLP [67], or others [83,88,119,127,136]) to assess their outcomes. Use of objective measures of activity level was less common (e.g., activity monitor—[115,123]), whereas two others used both (e.g., self-report and pedometer [112], self-report and exercise task in a lab [75]). Surprisingly, one study described using both self-report and pedometer recordings, but the authors did not report the pedometer results [136].

Not all of the PA and exercise outcomes were improved by these interventions. For example, one study [75] improved leisure PA (self-report), but not aerobic fitness (objectively measured). Another study [118] failed to demonstrate improvement in aerobic fitness (objective measure of maximum oxygen uptake) but reported improvements in muscle strength (objectively measured with dynamometer). Kcal burnt per week were improved in one study [112] as well as minutes of exercise per week, but no improvements were observed in number of steps per day. Another study reported significant change in minutes spent sitting per day, but not in the MET scores [83]. 

Of the 11 studies reporting improvements in PA and/or sedentary behaviour, three were RCTs [75,115,118], one was a non-randomised controlled study [67] and seven were uncontrolled studies [83,88,112,119,123,127,136]. Successful interventions (even if only for some outcomes) used the following types of PA interventions: (i) worksite intervention (incl. workstation treadmill, Wii system, short video clips with energetic activities, walking meetings) with health coaching via text messaging [115], (ii) healthy lifestyle website with discussion board [67], (iii) healthy lifestyle education group sessions, website, eHealth journal, Wii system at work [112], (iv) workstation wellness intervention (to increase standing, stretching and sipping water) [119], (v) minimal-contact self-managed (setting one’s own PA goal) pedometer program [123], (vi) workplace 1h/week light group exercise plus healthy lifestyle education classes [75], (vii) pedometer challenge [136], (plus recording daily steps on a website, with 10k daily steps goal) with educational classes on healthy lifestyle [83], or physical resources (water bottle, cookbook, prizes [127]), or both [88] also with extra group exercise sessions), or (viii) aerobics and resistance exercise with or without supervision. Several of the studies reporting improvements in PA included digital components to their intervention [67,112,115,123,136].

However, two other studies reported no improvement in PA, exercise or sedentary behaviour (one RCT: [76], one non-randomised controlled study: [96]) and one study reported decreased PA (daily steps and moderate-to-vigorous PA, both measured objectively) after the intervention (a non-controlled study: [113]). These unsuccessful interventions were based on: (i) worksite intervention (like in the study of [115]; also measured with activity monitor—[96], (ii) individual physiotherapy exercises with educational sessions [76]; however, this study compared the results between two groups that underwent physiotherapy exercises, with the only difference being extra educational sessions), (iii) a complex intervention (for increasing PA and diet, [113], including app for sharing recipes/tips/PA goals, Facebook groups for support, pedometer to set and monitor PA goals).

Two studies reported reductions in sedentary behaviour (e.g., minutes or hours spent sitting) [83,115]. One of these studies was RCT [115], one was an uncontrolled study [83].

##### Smoking

Only three studies reported smoking behaviour as an outcome [127,142,143]. Studies with smoking behaviour as an outcome included smoking education-based interventions [142,143], or PA combined with healthy eating education [127]. The two interventions that used education reported significant and positive effects on smoking behaviour (i.e., number of cigarettes smoked, number of people who stopped smoking, behaviour change stage). These two studies included one non-randomised controlled study [142] and one uncontrolled study [143]. The intervention that did not demonstrate a significant change in smoking behaviour was an uncontrolled study, which was focused on PA and healthy eating education [127]. 

#### 3.6.2. Psychological Health Outcomes

##### Stress and Coping

Stress (including stress-coping abilities) was the most frequently assessed outcome across the included studies (measured by self-report and physiological markers), being measured in 66 studies. There were 49 studies that reported improvements in a measure of stress [55,66,70,71,80,81,82,85,86,92,95,99,106,107,109,110,122,127,131,132,133,143,146,147,148,149,150,153,154,156,157,159,161,163,164,165,167,170,175,178,180,181,182,183,184,185,186,187,188]. Of the 49 studies reporting improvements in the measure of stress, 16 were RCTs [55,92,122,131,150,157,161,165,167,170,175,178,182,183,184,185], seven were non-randomised controlled studies [86,99,106,110,147,149,188] and 26 were uncontrolled studies [66,70,71,80,81,82,85,95,107,109,127,132,133,143,146,148,153,154,156,159,163,164,180,181,186,187]. The interventions in studies with positive outcomes used mindfulness/meditation [66,133,146,147,148,149,150,153,154,156,157,159,161,163,164,165], various forms of stress and coping education [55,66,70,71,80,81,82,85,86,92,95,99,106,107,109,110,143,153,156], alternative therapies (touch therapy/Reiki [181,182], light therapy [186], auriculotherapy [183,184,185], mantram repetition [187], NLP [188]), relaxation (e.g., resting with music [167], guided imaging [70,71], massage [170,175]), knitting [180]), low-intensity PA (yoga [122,132,133]), walking ([127,131]), and stress-inoculation training [178].

Another 17 studies [59,61,83,90,91,104,112,121,130,135,145,155,160,168,169,171,190] did not report any improvements in stress. Of these 17 studies, seven were RCTs [59,91,130,135,155,169,190], four were non-randomised controlled studies [61,145,160,168] and six were uncontrolled studies [83,90,104,112,121,171]). Two of these studies reported an increase in stress following intervention [59,90]. These two studies included one RCT [59] and one non-controlled study [90] with applied cognitive-behavioural therapy (CBT) and narrative training as their modes of intervention. The interventions that were not successful in improving measures of stress used multimodal interventions, PA, education, mindfulness, and relaxation-based (art, massage) interventions.

##### Depression and Anxiety

Thirty-two studies measured depression and/or anxiety as an outcome. A significant decrease in depression and/or anxiety was reported in 19 studies (six studies reported decrease in solely depressive symptoms [60,93,97,118,139,177], seven in solely anxiety [164,169,171,174,178,182,187], and six in both depression and anxiety [95,143,156,157,172,176]). Of the 19 studies reporting improvements in depression and/or anxiety, 10 were RCTs [60,118,139,157,169,172,174,176,178,182], two were controlled studies [93,177] and seven were uncontrolled studies [95,97,143,156,164,171,187]. In two studies measuring both depressive symptoms and anxiety [97,139], depressive symptoms improved but anxiety did not.

Studies that improved both outcomes used mindfulness-based interventions [156,157], education-based interventions [95,143], and various relaxation methods (feet bath [172], music, [172], music-based relaxation [176]). Studies that improved depressive symptoms used education with PA (aerobics, [60]), PA with supervision (aerobics and resistance, [118]), Omega-acid pills [139], mindfulness and compassion fatigue (CF) education [93], solely education (on self-care [97]), and cognitive-behavioural sleep intervention with listening to relaxing audio before bed [177]. The studies that improved anxiety used touch therapy [182], mantram repetition [187], full back massage [169], aromatherapy chair massage [171], meditation [164], Benson’s relaxation technique [174], or stress inoculation computer-based training [178].

An additional 13 studies found no significant intervention effect on these mood-related states [55,59,62,66,91,110,121,127,146,149,155,168,190]. Of the 13 studies reporting no changes in measures of depression or anxiety, six were RCTs [55,59,62,91,155,190], three were controlled studies [110,149,168] and four were uncontrolled studies [66,121,127,146]. Interventions that did not report any changes in depression and/or anxiety included stress education [55,91,110], SOC education with mindfulness [62], CBT education [59], compassion fatigue education with mindfulness [66], solely mindfulness [146,149,155], brief workplace PA [121], pedometer and healthy eating resources [127], relaxation (art, [168]) and workplace health screening (for work functioning impairments and mental health complaints, [190]).

##### Burnout and Compassion Fatigue

Thirty-five studies assessed burnout/compassion fatigue (CF) as the outcome. Of these, 21 found an improvement in all (13 studies; [63,65,66,70,71,93,105,133,149,152,153,162,180]) or certain subscales of burnout/CF (seven studies; [56,69,72,89,94,125,144]; EE in six, DP in four, PAch in one). Of the 21 studies reporting improvements in a measure of burnout/CF, four were RCTs [56,65,94,125], four were controlled studies [84,89,93,149] and 13 were uncontrolled studies [63,66,69,70,71,72,105,133,144,152,153,162,180]. The successful interventions relied on relaxation [63,180], education [56,65,84,89,94], mindfulness/ meditation [149,152], yoga [125] or multi-component interventions (e.g., mindfulness/PA—[133,144], mindfulness/education—[66,93,153], relaxation/education—[70,71,72,105], relaxation/meditation—[162], mindfulness/PA/education/relaxation—[69]). 

However, 13 studies did not find any improvement in measures of burnout/CF [59,80,81,82,121,139,145,148,151,155,158,161,179], with one reporting increased burnout [84]. Of these 14 studies, four were RCTs [59,139,155,161], five were controlled studies [84,145,151,158,179] and five were uncontrolled studies [80,81,82,121,148]. The one study (controlled design) that reported an increase in burnout after the intervention (albeit a smaller increase than in the control group) applied stress education [84]. Those interventions that were not successful in generating changes in measures of burnout applied mindfulness/meditation [145,148,155,158,161], mindfulness/relaxation [151], CF education [80,81,82], CBT education [59], brief workplace (10-min) PA [121], diet supplementation [139] or music relaxation [179].

##### Mindfulness

Only five studies reported an improvement in mindfulness following intervention [125,149,153,156,158]. Of these, one used RCT design [125], two were controlled studies [149,158] and two were uncontrolled studies [153,156]. A further four studies showed no improvements in mindfulness [91,145,148,163]. Of the four studies reporting no improvements in mindfulness, one was an RCT [91], one was a controlled study [145] and two were uncontrolled studies [148,163]. The studies that showed positive effects relied on various forms of mindfulness training [149,158], or mindfulness training as part of the intervention (with education [153,156], or yoga [125]). The studies reporting no changes in mindfulness reported interventions based on mindfulness [145,148], meditation [163] or stress education [91].

##### General Wellbeing and Life Satisfaction

Fourteen studies reported improvements in one or more outcomes related to general wellbeing (e.g., a measure of wellbeing, happiness, quality of life (QoL) and/or life satisfaction). Of these, three studies reported improvements in some measures of wellbeing but not others [161,168,187], whereas 11 studies reported improvements in all their included measures of wellbeing (n = 11 [62,65,67,74,76,102,147,149,151,156,183]). Of the 14 studies reporting improvements in general wellbeing and/or life satisfaction, five were RCTs [62,65,76,161,183], seven were controlled studies [67,74,102,147,149,151,168] and two were uncontrolled studies [156,187]. The successful interventions focused on solely meditation/mindfulness [147,149,151,161], mindfulness with SOC education [62], mindfulness with cognitive therapy [156], stress/coping education [65,74,102] with physiotherapy [76], healthy lifestyle website [67], and more unconventional methods such as mantram repetition [187], relaxation (silk painting, [168]) and auriculotherapy [183].

Conversely, nine studies [57,75,79,87,108,126,130,135,152] reported no significant improvements in measures of general wellbeing and/or life satisfaction. Of these, six were RCTs [57,75,79,108,130,135], one was a controlled study [126] and two were uncontrolled studies [87,152]. Interventions that did not change their outcomes most commonly relied on education (SOC model—[57], positive psychology—[79], searching for meaning—[108]), followed by mindfulness [152], mindfulness with PA [79,135], meditation with education about care for dying [87] and PA [130], or PA with stress/nutrition education [75]. The outcomes that did not improve were mostly measured in terms of life satisfaction/QoL [75,87,108,135], followed by measures of spiritual wellbeing [79], time being happy [79], and mental health [57,130,152]. 

##### Self-Efficacy

Only three studies reported self-efficacy as an outcome [58,124,150]. Improvements in self-efficacy (SE) were reported in two RCTs ([58]–self SE, [124]–exercise SE). The successful interventions used ergonomic education [58] and stretching PA [124]. One other RCT, with mindfulness intervention, reported no improvement in SE ([150]–caring SE).

#### 3.6.3. Work-Related or Organisational Outcomes

Thirty-two studies included at least one work-related or organisational outcome measure. Sixteen studies [68,73,76,87,92,95,103,104,108,120,130,135,139,160,179,190] showed positive effects on work-related outcomes, such as productivity and work ability [76,95,104,130,160,190], patient moving and handling procedures [68], sickness absence [130,160], presenteeism [139], management skills [73], workplace social capital [120], work satisfaction and/or attitude towards colleagues [87,135], work engagement ([92], measured by UWES-9), quality of work-life balance [103], team building ([179], as measured by self-report), work fatigue [104], and perception of work benefits [108]. Of the 16 studies reporting improvements in work-related or organisational outcomes, only eight were RCTs [76,92,108,120,130,135,139,190], three were controlled studies [103,160,179] and five were uncontrolled studies [68,73,87,95,104]. Interventions resulting in improved work-related outcomes used: (i) some form of education (improved productivity, better patient care, management skills, work-life balance, seeing benefits of work) [68,73,92,95,103,104,108], (ii) dietary supplements (reduced presenteeism) [139], (iii) music group sessions (improved team building) [179], (iv) meditation with relaxation/education (improved productivity, sickness days, work engagement, job satisfaction) [87,160], (v) physical activity (improved job satisfaction, work ability, work social capital) and/or education/mindfulness [76,120,130,135], and (vi) workplace health screening (improved work functioning [190]). 

A further 16 studies [55,57,59,62,82,84,111,112,115,144,148,150,161,165,171,189] did not report significant improvements in any measures of work-related or organisational outcomes. Of these, nine were RCTs [55,57,59,62,111,115,150,161,165], one was a controlled study [84] and six were uncontrolled studies [82,112,144,148,171,189]. These studies found no positive changes in job satisfaction [55,82,112,144,148,161,165], work functioning/productivity [57,62,111,115], caring efficacy [150,189], work-family conflict [59], sick days [75,115,171], job control ([57,62], as measured by the Work Analysis Instrument for Hospitals—Self-Report Version), work limitations ([55] as measured by the work limitations questionnaire) or work situation ([84] including job control, job demands and participation in decision-making). Interventions that did not show improvements in work-related outcomes were based on PA [115], meditation/mindfulness [148,150,161,165], mindfulness with education or exercise [62,144], solely education [55,57,82,84,112], education with workplace health screening [111], relaxation (massage [171]) and support groups [189]. 

None of the included studies reported intervention outcomes regarding early retirement/intentions, staff retention rates, or staff turnover rates. 

### 3.7. Success Rate of the Interventions

The success rate of interventions in improving the outcomes presented above is displayed in Table 2, as a percentage of included studies that measured each outcome. The highest success rate was for diet and nutrition interventions, followed closely by body composition, PA and stress/coping (all with above 70% of studies reporting at least some improvement). However, the evidence stemming from RCTs only is not clear for majority of the outcomes. Only body composition and stress coping seem to have strong RCT-based evidence for their effectiveness (in bold, Table 2).

### 3.8. Other Intervention Effects

Some outcomes were explored by only one or two studies. These included self-actualisation [67], rumination [159], obsessive passion [66], experiential avoidance [149], gratitude [79,95], risky driving [190], empathy [133], forgiveness [166], altruistic actions [166], compassionate love [166], positive outlook [95], resentfulness [95], marital satisfaction [101], beliefs about physiotherapy [129], seeking therapy [68], and serenity [133]. 

#### 3.8.1. Unintended Intervention Effects

Only ten studies (7%) reported no significant improvements in any measured outcomes following intervention [57,59,64,84,91,111,121,145,155,189]. Of these, two reported unexpected negative intervention effects, specifically, increases in burnout (EE, DP) [84] and stress [59]. Three other studies found significant improvements to some of their measured outcomes, but alongside positive outcomes they reported negative outcomes on other measures, such as increases in stress [90], a decrease in PA [113] and a decrease in emotional intelligence [73]. 

#### 3.8.2. Interventions with No Significant Positive Effect

Of the 10 studies that did not report a positive change in any outcome measure following intervention, five were RCTs [57,59,91,111,155], three were controlled studies [64,84,145] and two were uncontrolled studies [121,189]. These studies are briefly described below. Noben and colleagues [111] (n = 538) reported an RCT looking at the effects of occupational health screening on work functioning. They compared outcomes between three groups; screening with referral to a physician, screening with referral to e-health resources, and screening with no feedback. All screening conditions showed improved work functioning, although there were no significant differences between groups. Müller et al. [57] (n = 46; RCT) also reported no significant differences between groups (group with education based on SOC model vs. wait-list control group) in wellbeing, work ability and job control. Menzel and Robinson [59] (n = 20; RCT) compared CBT (focused on stress and pain management) with a wait-list control. These authors reported a non-significant trend (p = 0.06) towards pain reduction together with an unexpected significant increase in stress in the CBT group. There were no significant effects for mindfulness, burnout, or stress in a study by Horner and colleagues [145] (n = 43, pre-post controlled) when comparing mindfulness training with a passive control. Hartvigsen et al. [64] (n = 255, pre-post controlled) found no significant change in lower back pain when comparing a 2-year education intervention (body mechanics, lifting techniques) with a group that attended a single instructional meeting. Similarly, Freitas et al. [121] (n = 21, non-controlled) described no significant quantitative changes in anxiety, depression, burnout or job stress when comparing pre and post scores of a group that attended a 10-min PA workplace intervention, five times a week, for three months. There were also no significant changes in outcomes measured in a study conducted by Chesak et al. [91] (n = 40, pre-post controlled), where stress, mindfulness, anxiety and resilience were compared between a group that attended two education meetings, and a group that attended a single lecture on stress. No positive effects for burnout, depression or stress were reported in a study that compared an eight-week mindfulness course with a passive control [155] (n = 45; RCT). Also, a study reporting the effects of telephone support groups on stress, coping, job satisfaction and burnout, demonstrated no significant changes in outcome measures in this pre-post non-controlled study [189] (study 2; n = 15). Lastly, Le Blanc et al. [84] (n = 304, pre-post controlled) reported an increase in burnout when compared to baseline, after group sessions devoted to forming stress reduction plans within a nursing team. It should be highlighted that this increase in burnout was smaller than that reported by the control group.

A closer analysis of these studies suggests that null findings may have occurred due to the lack of complete data sets (as can be seen above), or due to a degree of similarity between intervention and control groups [111]. The only study for which this was not the case is the study of Hartvigsen and colleagues [64]. Their results may be partly explained by, in the words of the authors themselves, the fact that “the large number of teaching sessions may have increased awareness of back problems and in fact augmented the problem in the intervention group” (p. 16).

### 3.9. Dropout Rates

Of all the included studies, 17 (of which seven were RCTs) did not provide clear information on dropout rate. The remaining studies reported attrition rates ranging from 0% to 75%, with a mean of 18% (SD = 16%). The study with the highest dropout rate [61] used a 3-month follow-up, where the questionnaires were left in a staff room for two months, and thus problems with matching data occurred. The mean dropout rate shows that on average, data collection was completed with 82% of participants, thus results were not likely to have been strongly affected by attrition bias. The five studies that reported the highest attrition rates (>50%) were relatively long-term interventions (i.e., multiple sessions over multiple weeks/months; focused on stress coping, aromatherapy massage, or workplace PA; [94,121,171]), were based on one long session with no refresher sessions (e.g., development of self-care plan, [61]), or on two long sessions (i.e., learning stress symptoms and coping methods [95]).

### 3.10. Results of Included RCTs

In order to provide the most rigorous evidence, we also looked separately at the findings reported by RCTs. There were 52 standard RCTs (plus two cross-over designs), which included predominantly female and middle-aged nurses (M = 89%; M age = 37.70; SD = 6.30), with an average of 167 participants (mostly nurses; min = 14, max = 3381), and reporting an average dropout of 21% (SD = 18%; min = 0%, max = 75%). Thirty-five of the RCTs (67%) had a control comparison that received no intervention. These were described as control groups with no intervention (n = 21), wait-list comparison groups (n = 10) and usual care (n = 4). Nineteen studies provided some form of active intervention. Interventions used in these RCT studies were based on education (mostly stress coping, n = 15) [55,56,57,58,59,65,77,79,91,92,94,100,101,108,111], PA (n = 10) [115,118,120,122,124,125,128,129,130,131], relaxation (n = 8) [167,169,170,172,173,174,175,176], meditation/mindfulness (n = 6) [150,155,157,161,165,166], alternative approaches (n = 5) [182,183,184,185,190], diet (n = 2) [139,140], smoking cessation (n = 1) [141], and finally seven used multi-component complex interventions that included more than one element (e.g., PA with education, education with mindfulness) [60,62,75,76,135,138,178].

Results on the effectiveness of these interventions are presented in Table 3 (only results relevant to the aims of this SR are presented). In short, even amongst RCTs, results are rather mixed, with studies utilising the same type of intervention often reporting contrasting results. Some of the intervention studies report improvements in emotional (e.g., anxiety, depressive symptoms, stress, etc.) or physical (pain, muscle flexibility, strength) outcomes. However, improvements in work-related or organisational outcomes are less common, and the majority of RCT studies including these outcome measures show no effects. 

### 3.11. Risk of Bias Results

All the included studies (n = 136) were independently assessed for risk of bias by two reviewers (NSt, HB) with an initial agreement rate of 97.7%. A third reviewer (EK) independently assessed a 25% subsample. Disagreements were resolved by discussion between the reviewers to reach a consensus (i.e., where there was a disagreement the reviewers referred to the Cochrane Handbook, especially the definitions and examples for the bias assessment; and agreed an outcome that the most closely matched that guide). The Cochrane Handbook classification guide was followed, with reviewers assigning high, unclear or low risk level to studies in terms of six types of bias: (i) selection (random sequence generation, allocation concealment), (ii) performance (blinding of participants and personnel), (iii) detection (blinding of outcome assessment), (iv) attrition (incomplete outcome data), (v) reporting (selective reporting) and (vi) other bias.

The ‘other bias’ category was predominantly utilised in the current project to judge the adequateness of the sample size (here, n = 30 per condition was used as an adequate size threshold; as suggested by other authors e.g., [191,192]. It was also used to judge other aspects that may have influenced the data (such as contamination between conditions, etc.).

The results of the risk of bias analysis for all studies are displayed in Figure 2. In single group studies, blinding, randomisation and allocation concealment is not possible and therefore these studies were assessed to be at a high risk of bias in these categories. Amongst all the included studies, the highest proportion of bias was related to insufficient blinding of participants and/or personnel (111/136 studies). Other risks included lack of random sequence generation (74/136 high risk), insufficient or no allocation concealment (72/136 high risk), lack of blinding of outcome assessment (68/136 high risk), other sources of bias (64/136 high risk) and incomplete outcome data (52/136 high risk). The lowest proportion of studies with a high risk of bias was recorded for selective reporting bias (129/136 low risk). A significant proportion of studies did not adequately describe the process for collection of outcome assessment, resulting in unclear risk of bias for 54/136 studies. A high number of studies had limited reporting of allocation concealment and random sequence generation. Across all ratings, approximately 37% (352/945) of all risk ratings were low (46.6% was high, 16.4% unclear). Due to the high number of single group studies included in the review, across the seven categories of bias used only 17 studies (12.6%) were able to fulfil five or more low risk ratings, whereas only three studies (2%) reported blinding of both the personnel/participants and outcome assessment. It needs noting, however, that there is a high percentage of non-controlled one group studies in the current systematic review, which affects the risk of bias results, as for such studies blinding, randomisation and allocation concealment is not possible, and thus was assessed as presenting high risk. 

### 3.12. Quality Assessment

All studies were evaluated for methodological quality, with the use of CONSORT [52] (for RCTs) or TREND [193] (for non-randomised studies) checklists (see Appendix A). Quality varied across the included studies, with the lowest score of 7.5 (out of 23) being evaluated for a quasi RCT [101], followed by 8.5 (out of 20) for a controlled study [102], and 8 (out of 18) for a pre-post uncontrolled study [180]. There were only two studies that achieved the highest possible quality rating (both were RCTs: [62,138]); none of the pre-post controlled or the pre-post non-controlled achieved the full quality score. On average, the pre-post non-controlled studies scored 12.96 quality points, whereas the controlled studies scored 14.18. In comparison, RCTs earned on average 14.50 points. This suggests that the current literature has a high proportion of studies with low quality reporting, although there is a small number of publications that can be used as a reference point for reporting style. 

## 4. Discussion

This systematic review aimed to synthesise a substantial pool of evidence on the effects of lifestyle interventions on the physical and mental health of nurses, in addition to work-related outcomes. A total of 136 relevant studies were identified involving 16,129 participants who met all the inclusion criteria and none of the exclusion criteria.

### 4.1. Summary of the Characteristics of the Studies

The studies took place predominantly in hospital workplace settings and reported interventions that were based on (in the order of popularity): (i) various forms of education, (ii) physical activity, (iii) mindfulness/meditation, (iv) relaxation, (v) smoking, and (vi) other alternative non-medical approaches (e.g., art or alternative therapies). Interventions targeted various outcomes related to (in the order of popularity): (i) stress (66 studies), (ii) burnout/compassion fatigue [145], (iii) depression/anxiety [63], (iv) work-related outcomes [63], (v) body composition [118], (vi) wellbeing/QoL [182], (vii) physical health [57], (viii) PA [117], (ix) mindfulness [100], (x) diet and nutrition [100], (xi) self-efficacy [56], and (xii) smoking behaviour [56]. The average data completion rate of the included studies was 82%. Previous reviews with similar samples fail to report adherence rates (e.g., [194]); however, reviews of workplace interventions for employees suggest that this level of adherence is reasonable [195].

### 4.2. Successfulness of Interventions

The interventions were typically deemed to be more successful in relation to nutrition-related outcomes (88.9% of studies, 8/9 including nutrition as an outcome reported positive effects), followed by body composition (83.3%, 20/24), physical activity (78.6%, 11/14), and stress (74.2%, 49/66). Then smoking behaviour and self-efficacy (both with 66.7%, 2/3), physical health (64.7%, 11/17), wellbeing/QoL (60.9%, 14/23), burnout/CF (60.0%, 21/35), and depression/anxiety (59.4%, 19/32) were also successful, but to a lower degree. The least successful were interventions regarding mindfulness (55.6%, 5/9), and work-related outcomes, with the latter successful in only 50% of cases (16/32).

The review suggests that interventions aimed at improving nutrition amongst nurses commonly result in improved outcomes, especially when the interventions are education-based. However, it is important to note that the total number of studies including nutrition outcomes was very limited (only nine studies, with only two RCTs). This was not the case, however, for studies measuring body composition outcomes, which also had a high level of success, but more studies in this category utilised an RCT design. Of these, 10 RCTs showed improvement, and none of the RCTs showed no improvement, which provides a particularly clear and promising finding for the influence of lifestyle interventions on indices of body composition. Similarly, stress was measured in a high number of RCTs (22), with 16 showing significant improvements on this outcome. This also supports the credibility of lifestyle interventions for reducing stress in nurses. Physical activity was also a somewhat successfully improved outcome, with four RCT studies included in that category, including three that showed improved PA outcomes. Based on the above, we suggest that there is sufficient evidence to recommend the application of lifestyle interventions targeting body composition, stress, diet, and PA. However, more RCTs are required to provide additional higher quality evidence, particularly for diet and PA.

We found some evidence for improvements in smoking behaviour (66.7%), self-efficacy (66.7%), physical health (64.7%), wellbeing/QoL (60.9%), burnout/CF (60.0%), and depression/anxiety (59.4%), although the evidence for these is not so strong. This is partly due to the lower success rate of the studies reporting on these outcomes (than for body composition, stress, diet, and PA), and also because many of the studies reporting on these outcomes had lower quality designs or demonstrated quite ambiguous RCT-based evidence. First, none of the studies reporting on smoking behaviour had an RCT design. Only three RCTs assessed self-efficacy but one of these studies showed no improvement. Similarly, in terms of physical health, wellbeing/QoL and burnout, the results provided by the RCTs were mixed (i.e., four RCTs reported improvement (RCT+), while three RCTs reported no improvement (RCT-) for physical health, for wellbeing/QoL five were RCT+, six were RCT-, whereas for burnout four were RCT+, and four were RCT-). This suggests that more high-quality research is needed measuring these outcomes, and we need to better understand what moderates the effectiveness of these interventions. Although depression/anxiety had a lower success rate across all studies measuring this outcome (59.4%), when RCTs only were considered, the findings were more promising since there were 10 RCTs reporting improvements in depression/anxiety, although six RCTs showed no change. 

Finally, it is even more difficult to clearly describe the impact of interventions targeting mindfulness and work-related factors. These were the two outcomes with the lowest success rate across all studies measuring these outcomes (55.6%, and 50.0% of respective studies reported improvements on at least one relevant measure). To add to this, the RCT-based results also provided very ambiguous findings, where mindfulness was improved in one RCT, but showed no change in another, whereas work-related outcomes improved in eight RCTs, but did not show any change in another nine RCTs. Further, the type of intervention leading to improvements in these outcomes cannot be delineated. This highlights the need to more closely consider specific intervention aspects and their efficacy within targeted samples. Future research could conduct a meta-analysis of a narrower range of interventions and outcomes in order to address these questions.

Work-related outcomes, mindfulness, depression/anxiety, burnout/CF, and wellbeing/quality of life constructs were those outcomes that appeared to be less amenable to change with lifestyle intervention. It is important to consider the potential explanations for this. One potential barrier to modifying these factors is that they are complex outcomes and are influenced by multiple factors that may be more challenging to control through workplace intervention. For instance, work-related outcomes are likely to be influenced by factors that are not being targeted in lifestyle interventions, like the organisational environment and specific job stressors [196] such as work context, demands, pressure, or the perception of one’s role at work, etc. One approach that produced positive results targeted empowerment, civility and trust in management [197], however it was focused on creating a supportive and empowering work-environment rather than making the individual more resilient.

Furthermore, mindfulness is a particular skill that requires intensive training to be improved. Studying mindfulness presents many issues (e.g., potential for an initial increase in distress, [198]), and so adapting brief mindfulness interventions to workplace settings brings further challenges [199]. Future studies will need to consider these aspects. Also, researchers might consider using a recently developed framework for reporting mindfulness-based interventions [200].

Similarly, depression/anxiety, as well as burnout/CF, especially when clinically significant, might require individual professional mental help or counselling to generate positive outcomes, rather than a workplace lifestyle intervention. This will likely explain why the lifestyle interventions reviewed here were less likely to produce positive improvements for these mental health outcomes. Previous work has suggested that for burnout and CF, changes in organisational culture might be particularly important [201]. It has also been shown that interventions incorporating both personal and organisational aspects have more long-term effects for burnout [202]. Additionally, there is systematic review evidence suggesting that counselling is effective in alleviating psychological problems related to work [203]. Nonetheless, more holistic approaches (incorporating reducing work-related risk factors for mental health, developing positive aspects of work and employees, and addressing mental health problems irrespective of their cause) have been recently advocated [204]. Such initiatives need exploring more, as burnout has been identified as a leading cause of work-related mental health issues (e.g., [202]).

Lastly, wellbeing and quality of life are complex multidimensional concepts, which have been acknowledged as being difficult to change (e.g., [205]). Such factors may take a significant length of time to change; thus, short-term modifications to one’s health or lifestyle behaviour may not have immediate effects upon an individual’s overall perception of their life or general wellbeing, as such behaviours may need to be sustained for much longer periods to influence the more fundamental nature of wellbeing and quality of life. The majority of studies measured outcomes immediately post-intervention and did not assess outcomes in the medium or longer term when any changes to these outcomes may be more likely to have taken effect. It might also be the case that nurses who work in a particularly demanding work environment (with long shifts, problems with understaffing and over-utilisation of the health systems) do not perceive small individual changes (e.g., to health behaviours or psychological health) as salient enough to improve their overall quality of life. Given that shift-working nurses report lower quality of life than the general population [206,207], it is not surprising that improving their quality of life might be difficult to change at an individual level, and might require more complex changes at the organisational level. It is also true that quality of life as a concept has often been misunderstood in healthcare research [208], which might have affected the results presented here. In summary, all five of these outcomes might benefit from complex interventions that take a more holistic approach and pay attention to the conceptualisation and measurement issues. The variability in the measurement scales that were used to assess these constructs (as presented in the results section) provides additional evidence, both for the lack of consistency in measurement approaches, and lack of a consensus as to how to best measure these outcomes.

### 4.3. Results Specific to RCTs

Despite the inclusion of both RCTs and non-randomised studies in this review, results from RCTs only did not vary considerably from the findings based on the wider spectrum of the evidence reviewed. Similar types of interventions resulted in improvements, or no effects. Likewise, results relating to organisational outcomes showed little amenability to change. It is important to highlight that education-based RCTs were the only intervention type not to show any significant improvements in any outcome, whilst smoking-focused RCTs demonstrated only short-term effects. It seems likely that the provision of education-only might be the least beneficial to nurses and their organisations, and studies reporting on interventions targeting smoking behaviour are too few to draw meaningful conclusions. This is in agreement with the psychological literature, which warns that merely possessing knowledge does not necessarily lead to change in behaviour (e.g., attitude-behaviour gap; e.g., [209,210,211,212]. Moreover, it corroborates other findings from the nursing literature, suggesting that nurses, despite their training, knowledge and skills in health promotion practice, often do not practice what they preach (e.g., [7,8,42]). Thus, it might be crucial, if relying on education-based interventions, to offer them within a more multimodal context, which also focuses on aspects of behaviour change.

### 4.4. Quality Concerns

Many of the included studies were assessed as presenting low methodological quality which may have limited their ability to uncover intervention effects. While there were many randomised controlled trials included in this review (RCTs; n = 54; two with crossover design), there was a higher number of non-randomised studies (n = 82) that had pre-post designs (n = 31 with a control group, n = 51 without a control group). It is difficult to determine the effectiveness of interventions that have been tested using non-randomised designs, and these studies by their nature had higher risk of bias. The methodological quality of the studies varied substantially (including those tested in RCT designs), and there was a high number of studies that did not report enough information to make an assessment of quality in certain areas (see Figure 2 e.g., ‘blinding outcome assessment’ was unclear in a large number of studies). Future studies should adhere to CONSORT guidelines when designing, running and reporting intervention studies.

The main methodological concerns observed in the included studies were (i) absence of a control group or inclusion of a ‘non-active’ comparator group, which may obscure the actual effectiveness levels of the interventions; (ii) drop-out rate, with some studies reporting very high attrition from the research study, which may limit the true effect of an intervention; (iii) use of voluntary and small samples; with many studies not being randomised, and/or having very limited numbers of participants. In addition, only a very small sample of studies looked into long-term effects (>6 months) of the interventions, which limits the interpretation of their effectiveness after the intervention period is over.

### 4.5. Review Limitations

Efforts were made to minimise limitations such as the inclusion of risk of bias analysis and presentation of the quality assessments of the included studies. There are still, however, certain limitations that need to be taken into account when interpreting the result presented. The review was limited to articles published in the English language. Searches were undertaken by a single researcher, although there were two researchers involved in the overall process. There is a possibility therefore that some relevant literature was missed due to human error, or due to its publication in a language other than English. Studies with non-controlled designs were included in the review and assessed using the same stringent criteria, which increases the proportion of studies assessed as having a higher risk of bias. The fact that only 54/136 intervention studies utilised an RCT design needs to be taken into consideration when interpreting our findings, although to account for this, we have presented results separately for RCTs. Nevertheless, given the lack of recent reviews on lifestyle interventions and therefore the unknown scope and quality of evidence in this area, it was deemed important to employ broad inclusion criteria to capture details of relevant intervention studies and highlight the vast number of studies published in this field with low quality research designs and reporting. Our search criteria generated a large number of articles reporting the outcomes of a diverse range of interventions. However, it is possible that some articles were missed where particular search terms were not in our criteria (e.g., we did not specify ‘back pain’ or ‘musculoskeletal’ interventions in our search terms, although the review identified some articles with interventions in this area). There may be scope for a review focusing specifically on musculoskeletal interventions in nurses and/or other healthcare professionals. It is possible that our results were affected by publication bias, as null findings are less likely to be published. Lastly, a meta-analytical approach was not considered feasible due to the exceptionally high heterogeneity of outcomes and intervention modes used in the included studies.

### 4.6. Future Directions

One of the main issues identified in this review is the length of the interventions and timing of the post-intervention measurement. As reported above, the most common time frame for interventions was two months. Even though this seems like a considerable length, the habit formation literature suggests that it can take many weeks of daily repetition to establish a habit (e.g., [213,214]), which may not be possible with interventions running on a one time per week basis. Whilst daily home practice was stipulated by many of the interventions, it is not always clear whether this home practice actually occurred. It might be worthwhile to design and test interventions that maximise the ‘dosage’ of intervention by offering additional resources such as support group elements, or mobile support or reminders, in addition to encouraging and recording daily behaviours. For many studies, due to a low quality of reporting, it was not possible to determine the influence of factors such as intervention fidelity (relating to engagement and delivery), adherence and actual versus intended dosage, or attrition. Many studies did not report any theoretical framework or model for the intervention and the majority of studies did not use formal reporting guidelines in their publications.

## 5. Conclusions

In summary, this systematic review provides a comprehensive synthesis of the literature investigating workplace lifestyle interventions aimed at improving individual physical and mental health, and/or organisational outcomes in working-age nurses. This review highlights that there are significant methodological limitations in the published literature, with low quality of reporting regarding mostly interventions and research processes. This needs to be addressed in future studies with the increased use of standardised tools and checklists to inform intervention design and reporting. Tentative conclusions are drawn from a vast pool of research with mixed designs, heterogeneity of outcome measures, with a significantly smaller pool of higher quality RCT evidence. Overall, this review suggests that workplace lifestyle interventions targeting nurses are likely to have positive effects on a range of individual health and lifestyle factors such as diet and nutrition, body composition, PA and job-related stress. Findings for mindfulness, wellbeing/QoL, burnout/CF, depression/anxiety and work-related outcomes are more mixed, and may require novel, or more complex organisational approaches. Similar work needs to be undertaken among other groups of healthcare professionals, such as medics, whose health may have direct implications for the healthcare of their patients.

## Figures and Tables

**Figure 1 ijerph-17-00017-f001:**
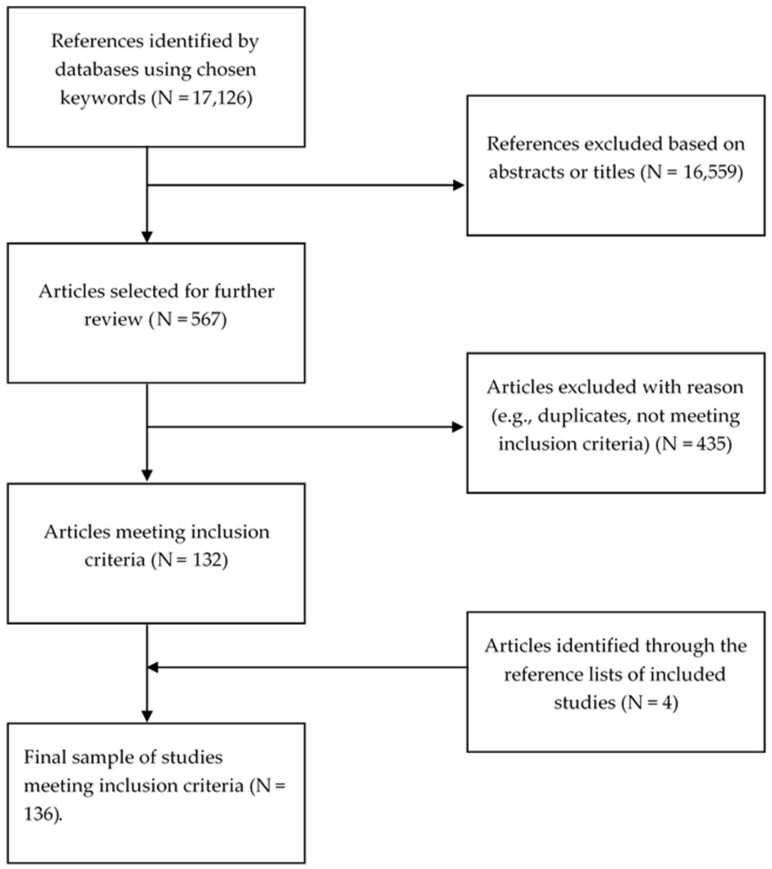
Flow diagram of study selection process (PRISMA).

**Figure 2 ijerph-17-00017-f002:**
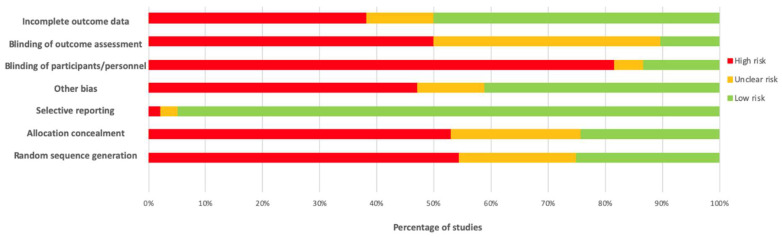
Estimated risk of bias across all studies (red = high risk, yellow = unclear risk, green = low risk).

**Table 1 ijerph-17-00017-t001:** The terms used in the search process.

Search Term 1–10	Search Term 11–20	Search Term 21–30	Search Term 31–40	Search Term 41–44
Physical Activity/	alcohol drinking or alcohol consum*. af	(hypertension or diabetes or coronary or renal failure or kidney failure or liver failure or cancer). af	mental health or mental illness or psychological or psychological wellbeing or stress or anxiety or burnout or depression or self-efficacy or self ADJ efficacy	(nurs$ not in-patient$ not inpatient$ not patient$).af.
Exercise/	1 or 2 or 3 or 4 or 5 or 6 or 7 or 8 or 9 or 10 or 11	13 or 14 or 15 or 16 or 17 or 18 or 19 or 20 or 21	23 or 24 or 25 or 26 or 27 or 28 or 29 or 31	12 or 22 or 32 or 39
(physical activity or exercise). af	Hypertension/	Mental Health/	Absenteeism/	40 and 41 and 42
Diet/	Diabetes Mellitus, Type 2/	Stress Disorders, Traumatic, Acute/	Job Satisfaction/	limit 43 to (human and English language) and yr = “2000-Current”
Obesity/	Acute Coronary Syndrome/	Anxiety/	Work Engagement/	
(diet or obesity or weight).af	Acute Kidney Injury/	Burnout, Professional/	Work Performance/	
Smoking/	Liver Failure/	Compassion Fatigue/	Patient Safety/	
Smoking Cessation/	Liver Failure, Acute/	Depression/	sicknessabsen* or absen* or job satisfaction or employee engagement or work performance or staff retention or staff turnover or patient safety or quality of care or patient care.af	
(smok*or cigarette$ or nicotine or tobacco).af	Peripheral Vascular Diseases/	Depressive Disorder/	33 or 34 or 35 or 36 or 37 or 38	
Alcohol Drinking/	Cerebrovascular Disorders/	Self-Efficacy/	(intervention or lifestyle or behavior* or behavior* change intervention* or behavior* change technique*).af.	

**Table 2 ijerph-17-00017-t002:** The success rate of the interventions in improving outcomes (in descending order).

Outcome	No. of Studies Reporting Improvement	No. of Studies Reporting the Outcome	Success Rate (% Reporting at Least Some Improvement)	No. of RCTs	
				RCT+	RCT−
Diet and nutrition	8	9	88.9	2	0
Body composition	20	24	83.3	**10**	**0**
PA	11	14	78.6	3	1
Stress and coping	49	66	74.2	**16**	**6**
Smoking cessation	2	3	66.7	0	0
Self-efficacy	2	3	66.7	2	1
Health and physical symptoms	11	17	64.7	4	3
Wellbeing and QoL	14	23	60.9	5	6
Burnout/CF	21	35	60.0	4	4
Depression and Anxiety	19	32	59.4	10	6
Mindfulness	5	9	55.6	1	1
Work-related	16	32	50.0	8	9

Note. RCT+(RCTs with improvement), RCT−(RCTs without improvement). In bold are outcomes with relatively clear results from the RCTs.

**Table 3 ijerph-17-00017-t003:** The results reported by RCTs, divided by their intervention focus.

Type of Intervention	No. of Studies	Improvements	No Effect	Comments
Education	15	Physical health [100], Emotional intelligence [100], Self-efficacy [58,101], Correct body posture [58], Stress [55,92], Spirituality [77], Gratitude [79], Work engagement [92], Emotional exhaustion [54,65,94], Depersonalisation [54,65], Personal achievement [65], Mental health knowledge [65], Marital satisfaction [101], Perception of work benefits [108]	Work functioning [111], Work ability [57], Work limitations [55], Mental health [57], Job control [57], Quality of life [108], Mood [59], Happiness [79], Job satisfaction [55,108], Coping [55], Resilience [91], Anxiety [91], Stress [91], Mindfulness [91], Depersonalisation [94], Personal achievement [54,94]	4 studies with no significant improvements in any result [57,59,91,111]; 1 reported increase in stress [59]
PA	10	Depressive symptoms [118], Muscle strength [118], Muscle flexibility [128,130], Metabolic indicators [118], Blood pressure [131], Work stress [122], Sleep quality [122], Pain [124,129], Exercise self-efficacy [124], Work social capital [120], Work ability [130], Mindfulness [125], Self-care [125], EE and DP [125], Body fat [131], PA [115], BMI [115]	Aerobic fitness [118], BMI [131], Waist size [131], Personal achievement [125], Physical health (incl. cardiovascular health) [129,130], Pain chronicity [129], Wellbeing [130], Work stress [130], General stress [130]; Work productivity [115]	
Relaxation	8	Stress [167,170,172,175], Heart rate [167], Blood pressure [167], Cortisol level [167], Anxiety [169,174,176], Depression [176], Mood [172], Psychosomatic symptoms [176], Sleep quality [173]	Blood pressure [169,172], Cortisol level [169]	6 studies reported only positive changes [167,170,173,174,175,176]
Meditation/mindfulness	6	Stress [150,157,161,165], Depression [157], Anxiety [157], Affect [165], Resilience [165], Wellbeing [161], Physical symptoms [157], Altruism and Perspective taking [166]	Job satisfaction [161,165], Burnout [155,161], Depression [155], Stress [155], Personal distress [166], Caring efficacy [150], Vitality [161]	1 study reported no positive changes [155]
Diet	2	Depressive symptoms [139], Insomnia [139], Presenteeism [139], Snack intake [140]	Anxiety [139], Burnout [139]	
Smoking	1	Abstinence rate [141]		Short-term only
Alternative	5	Stress [183,184,185], Work functioning [190], Mental health [183], Coping [184]	Distress [190], Depressive symptoms [190], Anxiety [190], Need for recovery after work [190]	1 study [182] reported improvements in anxiety, relaxation, and physiological state; but the same was true for mock intervention
Complex	7	Depressive symptoms [60], Anxiety [178], Pain [138], Coping skills [178], Muscle strength [138], Musculoskeletal complaints [75], PA [75], Work ability [76], Work wellbeing [76], Work satisfaction [135], Quality of life [62]	Aerobic fitness [75], Quality of life [75], Life satisfaction [135], Health complains [75], Muscle strength [75], Anxiety [62], Depressive symptoms [62], Stress [135], Work ability [62], Job control [62], Work absence [75,135]	3 studies reported only positive changes [60,138,178]

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
