# Peer review of "Effectiveness of Lifestyle Health Promotion Interventions for Nurses: A Systematic Review"

_ijerph, 2019, doi:10.3390/ijerph17010017_

Round 1

Reviewer 1 Report

Review of “Effectiveness of Lifestyle Health Promotion Interventions for Nurses: A Systematic Review”

Overall, this is a comprehensive necessary review and the authors did very well with this exhaustive review.

To make the presentation of the work more clear, please reformat the tables. How many of the lines do not line up may be confusing to some readers. This might leave more blank space, but there will be a “cleaner” presentation.

Line 198-199 (2.4. Data Extraction)- Can the authors clarify the process of agreement? “Reached through discussion” is moderately vague. Perhaps there was a flow chart/systematic process for how discussions took place? Or was it just free-form?

There seemed to be a spacing error at line 375 (MBI).

Line 446- Other Clinical Outcomes: Omit this section or just have a concluding sentence in the prior section.

Line 770- “Disagreements were resolved by discussion.” Like before this is a bit vague.

Author Response

Dear reviewer,

Thank you for providing your review. We are pleased to hear that our work was met with your positive feedback. 

We have taken every effort to address your comments, and thus (1) we have added a comment regarding the agreeing process between the authors, (2) we removed the spacing error, (3) we added the sentence mentioned below (line 446) to the paragraph presented above it, and (4) we added a comment about resolving disagreements for a better clarity. 

However, (5) the work on the  presentation of the tables seems to be done by the journal, as at the moment the tables do not appear in the manuscript that we were given by the editor.

Thank you for your suggestions, addressing them helped with improving the quality of our work.

Natalia Stanulewicz (corresponding author)

Reviewer 2 Report

I feel this is a very laborious piece of work that contributes substantially to the understanding the particulars of lifestyle promotion interventions in nurses.

One minor thing to add in the conclusions would be to suggest the comparison of such interventions with similar ones in other occupational groups such as medical doctors or other health professionals.

Author Response

Dear reviewer,

Thank you for providing your comments regarding our work, and for appreciating the amount of work that was done to prepare this manuscript. We really appreciate you commenting on that. 

In line with your suggestion, we have added to our conclusions the idea for similar work being undertaken with other occupational groups.

Thank you for spotting this omission on our part.

Natalia Stanulewicz (corresponding author)

Reviewer 3 Report

Dear Authors

The paper concerns an important current problem related to the health promotion methods and their effectiveness among nurses, who perform "helping profession". Health and well-being of nurses influence also well-being of the patients.

I find this paper most interesting. The manuscript is an overview and synthesis of the effectiveness of interventions conducted in nurses. This is a review of the papers published from January 2000 to December 2018. This is a long time which resulted in a large number of publications (17,126 papers as a result of the first identification). Authors did a huge job to find papers using correctly chosen key-words and selected them according to the proper criteria. They used PRISMA procedure. Selection of the key-words shows good knowledge of the problem. But here I have a question, why Authors didn't use a key-words "musculo-skeletal disorders- (MSD)" and "musculo-skeletal symptoms", "back pain" etc. However Authors in line 278 mentioned "prevention of back pain"

MSD are very popular ailments among nurses and health promotion addressed to these problems has been the subject of research. I think that this lack should be explained, maybe in part "Limitation" .

It is not clear if Authors analyzed only Original studies or also Reviews?

I have also some minor remarks:

In my opinion the table 1 is not clear, numbers should be near the particular key-words (terms), because it is difficult to assign the number to the appropriate term. It is not expalnation of asterix (*)- what it means? In publications it is assumed that if the abbreviation is used for the first time in the text, it should be explained. Authors used some abbreviations (RCTs, PA, QoL, CF, CBT, SOC) without providing a shortcut extension. Not for all readers it will be clear. In the Table 3 it is worth quoting the paper number in the columns "Improvements" and "No effects"

I suggest to change the part Key-results. Now, in my opinion there are not Key-results, but rather summary of the number of papers related to different kind of interventions.

Despite some comments I rate this paper as well done and certainly worth publishing.

Author Response

Dear reviewer,

Thank you for providing comments on our manuscript and the positive feedback, particularly noticing the substantial amount of work that was done to complete this project. We really appreciate this.

In response to your comments, 

(1) We have acknowledged the omission of the musculo-skeletal disorders in our limitations section, we did include back pain, but excluded disorders in this review, (2) we clarified that this review only includes original studies (see eligibility criteria), (3) we have made every effort to make sure that the abbreviations are explained in the text, when they are used first time, (4) we added the numbers of papers in Table 3, and (5) we changed "key results" to "summary of the characteristics of the studies".

In regard to table 1, following discussion with co-authors, we have chosen not to revise the search strategy table (Table 1) as it is not clear what is required.  The search strategy was developed in consultation with a senior university librarian and this is considered to be a standard way to present the search strategy in a systematic review with MeSH and free text terms. However, we added a note explaining the role of the asterisk.

We believe that these changes have improved our paper, so thank you once more for your helpful comments.

Natalia Stanulewicz

(corresponding author)